# A Phase II, Open-Label Clinical Trial of Intranasal Ketamine for Depression in Patients with Cancer Receiving Palliative Care (INKeD-PC Study)

**DOI:** 10.3390/cancers15020400

**Published:** 2023-01-07

**Authors:** Joshua D. Rosenblat, Froukje E. deVries, Zoe Doyle, Roger S. McIntyre, Gary Rodin, Camilla Zimmermann, Ernie Mak, Breffni Hannon, Christian Schulz-Quach, Aida Al Kindy, Zeal Patel, Madeline Li

**Affiliations:** 1University Health Network, University of Toronto, Toronto, ON M5S 2E8, Canada; 2Department of Psychiatry, Netherlands Cancer Institute, 1066 CX Amsterdam, The Netherlands

**Keywords:** ketamine, esketamine, NMDA, psychedelics, major depressive disorder, cancer, palliative care, antidepressant, end of life, supportive care, quality of life

## Abstract

**Simple Summary:**

Ketamine has demonstrated rapid antidepressant effects, but has been minimally studied in cancer populations. We conducted an open-label trial evaluating ketamine for depression in patients with advanced cancer. Participants received three flexible doses of intranasal (IN) ketamine (50–150 mg) over a one-week period. Twenty participants were enrolled in the trial, receiving at least one dose of IN ketamine. We observed rapid, robust and partially sustained antidepressant effects with flexibly dosed IN ketamine with adequate safety and tolerability in individuals with moderate to severe depression comorbid with advanced cancer. Given these promising findings, larger, controlled trials are merited.

**Abstract:**

Antidepressants require several weeks for the onset of action, a lag time that may exceed life expectancy in palliative care. Ketamine has demonstrated rapid antidepressant effects, but has been minimally studied in cancer and palliative care populations. Herein, the objective was to determine the feasibility, safety, tolerability and preliminary efficacy of intranasal racemic ketamine for major depressive disorder (MDD) in patients with advanced cancer. We conducted a single-arm, open-label phase II trial at the Princess Margaret Cancer Centre in Toronto, ON, Canada. Participants with advanced cancer with moderate to severe MDD received three flexible doses of intranasal (IN) ketamine (50–150 mg) over a one-week period. The primary efficacy outcome was an antidepressant response and remission rates as determined by the Montgomery–Åsberg Depression Rating Scale (MADRS) from baseline to the Day 8 primary endpoint. Twenty participants were enrolled in the trial, receiving at least one dose of IN ketamine, with fifteen participants receiving all three doses. The Day 8 antidepressant response (MADRS decreased by >50%) and remission (MADRS < 10 on Day 8) rates were high at 70% and 45%, respectively. Mean MADRS scores decreased significantly from baseline (mean MADRS of 31, standard deviation 7.6) to Day 8 (11 +/− 7.4) with an overall decrease of 20 points (*p* < 0.001). Antidepressant effects were partially sustained in the second week in the absence of additional ketamine doses, with a Day 14 mean MADRS score of 14 +/− 9.9. Common adverse effects included fatigue, dissociation, nausea, dysgeusia and headaches; almost all adverse effects were mild and transient, resolving within 2 h of each ketamine dose with one dropout related to adverse effects (negative dissociative episode). Given these promising findings, larger, controlled trials are merited.

## 1. Introduction

Major depressive disorder (MDD) in patients with advanced cancer is common, with an estimated prevalence of 10–20% [1,2]. Several meta-analyses have shown that MDD is an independent risk factor for poor prognosis, with reduced survival in patients with cancer [3,4,5]. Depressive symptoms are commonly associated with existential distress, death anxiety and increased pain in patients with advanced cancer [6,7,8]. Early and effective treatment of depression is essential to reduce end-of-life distress and improve the quality of life in patients with advanced cancer [9].

Monoaminergic antidepressants, such as selective serotonin reuptake inhibitors (SSRIs), require several weeks to achieve an antidepressant effect. Psychotherapy also takes several weeks to months for significant antidepressant effects to occur and may be further impeded by cancer-related fatigue and cognitive impairment [10]. For patients near the end of life, this lag time may exceed their life expectancy. Systematic reviews and meta-analyses have also demonstrated the limited benefits of antidepressants for MDD in cancer compared to the general population [11,12].

Taken together, there is a great need to identify a fast-acting and effective antidepressant for individuals with advanced cancer. Important aspects to consider in this population are ease of administration, portability (i.e., ideally a treatment that can be self-administered at home), safety and tolerability. One candidate that may fulfill these criteria is ketamine, given its robust and rapid antidepressant effects. Intranasal (IN) ketamine is of particular interest because of its portability and ease of administration, as compared to the intravenous (IV) route studied in the initial ketamine clinical trials [13,14].

While the antidepressant efficacy and safety of ketamine have been extensively studied in general psychiatric populations, minimal research has been conducted specifically in palliative care and cancer populations [13,14]. Of note, racemic ketamine is an off-label treatment for MDD, while esketamine nasal spray is FDA-approved as an add-on treatment for treatment-resistant MDD. Ketamine has been routinely and safely administered through a variety of routes (subcutaneous, IV, oral, rectal) for the management of pain in palliative care settings [15], with less research evaluating antidepressant effects in this population [16]. A few case reports, a case series and one small open-label trial suggest a positive effect of IV, subcutaneous (SC), intramuscular (IM) or oral administration of ketamine to ameliorate mood and anxiety symptoms in patients with advanced cancer [17,18,19,20,21]. However, to our knowledge, there are no reported studies evaluating IN ketamine for depression in palliative care or oncology populations. As such, our objective was to evaluate the feasibility, safety and preliminary efficacy of IN ketamine for moderate to severe MDD in individuals with cancer receiving palliative care.

## 2. Materials and Methods

This study was a single-center, 14-day, open-label, flexible-dose, single-arm clinical trial of IN racemic ketamine for moderate to severe MDD. This study was conducted at the Princess Margaret Cancer Centre (PM), University Health Network (UHN), in Toronto, Ontario, Canada. The study was approved by the University Health Network Research Ethics Board (REB; approval #16-5754) and Health Canada and registered on ClinicalTrials.gov (NCT03410446).

### 2.1. Eligibility and Recruitment

This study had a 3.5-year recruitment period (September 2018 to November 2021) to determine the merit and feasibility of a future RCT in the proposed setting and population based on the observed antidepressant efficacy, safety, tolerability, acceptability and recruitment/retention rates. The study was initially restricted to the inpatient palliative care unit; however, due to slow recruitment in the first year, the protocol was amended to include PM outpatients and patients admitted to any unit at PM. Eligibility criteria were the following: (1) Provide written, voluntary, informed consent prior to study enrollment; (2) Males and females ≥18 years of age; (3) Meet DSM-5 criteria for MDD, with a current major depressive episode (MDE). A diagnosis of a current MDE was confirmed using a mini international neuropsychiatric interview (MINI) conducted by a delegated member of the research personnel while assessing eligibility; (4) Depression severity in the moderate to severe range, as determined by a MADRS [22] score greater than or equal to 20; (5) Confirmed diagnosis of cancer and an estimated life expectancy of less than 12 months, as determined by the clinical care team; (6) No presence of delirium, suspected delirium or clinically significant confusion; (7) No severe hypertension (systolic blood pressure greater than or equal to 160 and/or diastolic blood pressure greater than or equal to 100) or severe cardiac decompensation; (8) No previous stroke history (based on history and review of medical records); (9) No history of intolerability, hypersensitivity or allergy to ketamine; (10) No history of bipolar disorder, psychotic disorders, substance use disorders or active suicidality based on a MINI conducted by a delegated member of the research personnel; (11) No current symptoms of psychosis or perceptual disturbances of any kind per investigator discretion; (12) Not a pregnant or breastfeeding woman.

### 2.2. Intervention

Treatment was administered on-site at PM (e.g., no at-home administration) in the inpatient units for admitted patients or in the research day treatment unit for outpatients. Ketamine was administered via nasal spray in prefilled syringes with 0.5 mL of sterile ketamine solution (Sandoz ketamine hydrochloride at 50 mg/mL, i.e., 25 mg per syringe) using the MAD^TM^ Nasal Intranasal Mucosal Atomization Device, as used in previous studies [23].

On Day 1 of the study, participants received their first dose of 50 mg intranasal ketamine. On Day 4 of the study, participants received a second dose of ketamine. Based on the tolerability and antidepressant effect of the first dose, the dose could be increased up to a maximum of 100 mg based on the discretion of the study investigator in shared decision making with the participant. On Day 7 of the study, participants received a third dose of ketamine. Based on the tolerability and antidepressant effects of the first and second dose, the dose could be increased up to 150 mg based on the discretion of the study investigator in shared decision making with the participant. No specific formal cut-offs or measures were used to determine adequate tolerability for a dose increase. General global tolerability was determined clinically by the study investigators and discussed with the participant for potential dose increase. For example, if a participant had minimal adverse effects and did not achieve remission with the 50 mg first dose, the dose may be increased to 100 mg for the second dose. If adverse effects were moderate, a dose increase to 75 mg would be considered to reduce the risk of severe adverse effects with the dose increase. If the participant achieved full remission at a given dose, the dose would not be increased, regardless of tolerability.

Participants were allowed to take concomitant psychiatric medications; however, no psychotropic medication changes could take place two weeks prior to or during the 14-day period of the trial. No specific psychotherapeutic interventions were included as patients were seen by consultation–liaison psychiatrists as part of routine care; however, there were no specific psychotherapeutic interventions as part of the trial (i.e., we did not provide ketamine-assisted psychotherapy). If a participant had an adverse reaction requiring the administration of psychotropic medications (e.g., antipsychotic medication for severe treatment-emergent perceptual disturbances), the on-call consulting psychiatrist could prescribe psychotropic medications at their clinical discretion; however, this was never required. Due to the medical complexities of this population, the protocol did not dictate specific supportive measures as treatments may have been highly variable depending on the medical condition and concomitant medications that each individual participant was receiving. All non-psychotropic medication changes were made by the attending palliative care staff physician as indicated for comorbid symptom management.

### 2.3. Outcome Measures

Medical variables to be used for descriptive purposes were extracted from the medical chart, including patient age, sex, cancer type and stage, estimated life expectancy (verified with the clinical team), previous psychiatric and medical diagnoses, current medications, medication changes and baseline emotional distress scores. Participants were followed for 14 days following the administration of the first dose of ketamine, regardless of whether or not they received the subsequent two doses. Tracking of effects and adverse effects was conducted daily. Given that the participants were medically unwell, study measures were selected to minimize the burden of participation by limiting the number of repeated measures and time required to obtain outcomes. Study measures were obtained by the research coordinators trained in the administration of all scales. The same study coordinator performed baseline and endpoint measures to allow for consistency in rating and to avoid potential biasing of results from inter-rater variability.

The primary efficacy outcome was the MADRS. The MADRS is a clinician-rated depression severity rating scale, specifically designed to be sensitive to change and independent of frequency of administration [22]. A clinician-rated scale was used because the Food and Drug Administration (FDA) recommends the use of validated clinician-rated depression scales due to improved reliability and sensitivity to change; the MADRS is an FDA-approved scale for use in depression trials [24]. As it is sensitive to change over a short time period, it has often been used in clinical trials assessing the rapid antidepressant effects of ketamine [14]. The minimal clinically important difference (MCID) in the MADRS score has been found to be 1.6 to 1.9 [25]. The mean change in MADRS scores 24 h after ketamine administration has been found to be 5–10 points in previous studies [14,26,27]. Validation in patients with cancer against structured clinical interviews is high [28,29]. Non-conflicted clinical research coordinators (CRCs), or study coordinators, were contracted from the PM clinical trials unit (third party outside of core study investigators) and were trained on the MADRS and regularly evaluated for inter-rater reliability. CRCs could confer with investigators about scoring; however, investigators avoided directly rating participants to reduce bias in over-estimating improvements in this open-label study.

Secondary outcomes included patient-reported outcome measures (PROMs) of depression, anxiety and pain captured through the Patient Health Questionnaire-9 (PHQ-9) [30], Generalized Anxiety Disorder-7 (GAD-7) [31] and the Edmonton Symptom Assessment Scale revised (ESAS-r) [32], respectively. This study utilized the Common Terminology Criteria for Adverse Events (CTCAE) v4.0 for toxicity and adverse event reporting (http://ctep.cancer.gov accessed on 1 December 2022), monitoring for adverse events on a daily basis during the 14 days of the clinical trial.

### 2.4. Statistical Analysis

The pre-set criteria to determine the merit of a future RCT were based on efficacy criteria standardly utilized for single-arm, open-label trials with the threshold for a positive proof of concept being set at >30% of participants achieving a clinical response (MADRS score decreasing by >50%) on Day 8 compared to baseline, similarly to previous feasibility studies [33,34]. In addition, we conducted two-tailed t-tests (alpha < 0.05 for statistical significance) to compare baseline to Day 8 (primary endpoint) for primary and secondary outcomes with an intention to treat (ITT) analysis using last observation carried forward (LOCF). All other time points used were secondary endpoints of interest to determine the rapidity and durability of antidepressant effects. Descriptive statistics on the adverse effect scales and recruitment/retention rates were also completed to evaluate the feasibility of a future RCT.

## 3. Results

### 3.1. Recruitment

The recruitment and enrollment summaries are shown in the CONSORT flow diagram in Figure 1. From 2018 to 2021, 33 patients were referred to the study for potential participation, with referrals and recruitment significantly reduced due to the global pandemic and research restrictions. Of the 33 patients referred, 22 were eligible and consented to participate, with 20 participants receiving at least one dose of ketamine (n = 15 received all three doses, n = 3 received two doses and n = 2 received one dose). One dropout was related to adverse effects of ketamine. The other four dropouts were unrelated to study participation, but were related to changes in medical status (i.e., discharge from hospital or progression of cancer complications preventing participation).

### 3.2. Baseline Demographics

Baseline demographic information for all participants receiving at least one dose of ketamine is summarized in Table 1. Of note, mean baseline depressive symptoms were in the moderate range at 31, ranging from moderate to severe. Baseline anxiety was also in the moderate range. Life expectancy ranged from 1 to 12 months and all included participants had stage 4 cancer. Baseline psychiatric medications and cancer types are summarized in Table 1.

### 3.3. Changes in Depressive Symptom Severity

As shown in Figure 2, mean MADRS scores (standard deviation) decreased from a baseline score of 31 (7.6) to 17 (8.0) on Day 2 after the first dose of ketamine. MADRS scores decreased further with each additional ketamine treatment to a mean score of 11 (7.4) by the Day 8 primary endpoint, 24 h after the third and final ketamine treatment; an overall mean reduction of 20 points on the MADRS was observed comparing the baseline to the Day 8 primary endpoint, in keeping with a clinically and statistically significant improvement (95% confidence interval: −24.7 to −15.3; *p* < 0.001). A reduction in MADRS scores was partially maintained in the second week, in the absence of additional ketamine treatments, with a Day 14 mean MADRS score of 14 (9.9).

Response and remission rates were also evaluated for MADRS on Day 8 (primary endpoint). A total of 14/20 (70%) participants met antidepressant response criteria (e.g., MADRS decreased by >50%) and 9/20 (45%) met remission criteria (MADRS < 10) by the Day 8 primary endpoint.

### 3.4. Secondary Outcomes

Changes in self-reported depressive symptoms (PHQ-9), anxiety symptoms (GAD-7) and pain (ESAS-r pain) are summarized in Figure 3. As shown, self-reported depression and anxiety symptoms were significantly reduced from baseline to Day 8 (*p* < 0.001 for PHQ-9 and GAD-7), whereas pain levels had a non-significant trend towards improvement post-treatment.

### 3.5. Safety and Adverse Events

Adverse events using the CTCAE v4.0 are reported herein as summarized in Table 2. Dysgeusia was the most common side effect (50%). Dizziness, fatigue, nausea, headaches and dissociation were other common adverse events. Only one participant dropped out of the study due to adverse effects (i.e., dissociation), while all of the other participants described adverse effects as being mild and transient, mostly resolving within 2 h of dose administration.

## 4. Discussion

The present study demonstrated rapid, robust and partially sustained antidepressant and anxiolytic effects with three flexible doses of IN racemic ketamine in participants with advanced cancer receiving palliative care. Ketamine was well tolerated in this medically ill population with adverse effects generally being transient and a low all-cause dropout rate in this pilot study. Preliminary efficacy, tolerability and safety support the merit of a future RCT to more rigorously evaluate this intervention and to determine sustainability of the effect. A feasibility evaluation suggested that the inclusion of both outpatients and inpatients would be required to have adequate recruitment rates. To improve feasibility, we also needed to expand to other inpatient units and broaden the life expectancy criteria from less than 6 months to less than 12 months. While recruitment was partially impacted by pandemic restrictions, the low recruitment rate also suggested that a multi-site study would be required for an adequately powered RCT to be successfully completed.

Reductions in MADRS scores were greater with repeated and increasing doses, suggesting repeated doses are likely beneficial for optimal benefits, rather than single doses as studied in the early ketamine trials [35,36]. Overall MADRS score reductions were greater than most ketamine trials conducted in general psychiatric populations [14,27]. This finding may be related to lower degrees of treatment resistance, open-label design or factors specific to cancer-related MDD, such as the upregulation of NMDA receptors in cancer, that could theoretically enhance ketamine’s biological effects as an NMDA antagonist [37]. In addition, ketamine has been found to be more effective in patients with increased inflammation [38,39], suggesting the possibility that it can alleviate inflammatory “sickness behaviours”, a constellation of symptoms including altered energy, appetite, motivation, sleep and cognitive function [40], in this population. Additionally, of note, the observed reduction in depressive symptoms was greater than reductions observed in previous studies using alternative routes of administration in this population (oral, SC, IM) [17,18,33]. It is not possible to directly compare these heterogeneous studies; however, our preliminary results suggest that IN may be a preferable route of administration that avoids invasive procedures while maintaining antidepressant efficacy with a large effect size.

The present study also adds to the limited research on IN racemic ketamine, suggesting that it might have efficacy and safety similar to IN esketamine that is approved for treatment-resistant MDD [27]. Off-patent racemic ketamine may allow for improved cost-effectiveness due to the high cost associated with the patented IN esketamine formulation. Additionally, the present study adds to the scant literature on potential anxiolytic effects of ketamine, with large decreases in anxiety symptoms observed.

The present study has several important limitations. Most notably, as a single-arm, open-label study, it is not possible to account for placebo effects, regression to the mean and expectancy bias. More recently, given the widespread enthusiasm for psychedelics in palliative care, expectancy bias may have been further enhanced [41]. Of note, our results were comparable to the very large effect sizes observed with psilocybin-assisted therapy for depression and anxiety in cancer, palliative and end-of-life populations [42]. The small sample size is another major limitation. A relatively short follow-up period also prevents adequate determination of the durability of antidepressant effects. This study was also unable to answer key questions about treatment optimization, such as defining the optimal dosing regimen and the utility of combining ketamine with psychotherapy.

Given these important limitations, and despite our promising results, we believe that ketamine should not be considered to be a replacement for conventional treatment and psychological support. Indeed, the importance of psychological interventions in advanced cancer, palliative care and near the end of life is strongly emphasized as a vital part of cancer care [4].

## 5. Conclusions

In conclusion, the present pilot trial shows promising results for IN racemic ketamine for MDD in advanced cancer. There is a great need for rapid-acting, effective and safe antidepressant treatments, given the limitations of currently available treatments and the negative impact of MDD on patient outcomes in those with an advanced disease. Our results provide sufficient evidence to support the merit and feasibility of a future RCT to more rigorously evaluate safety and efficacy of this intervention. As recruitment was the main limiting factor of study feasibility, including outpatients and inpatients in a multi-site trial will be critical to allow for the completion of an adequately powered RCT.

## Figures and Tables

**Figure 1 cancers-15-00400-f001:**
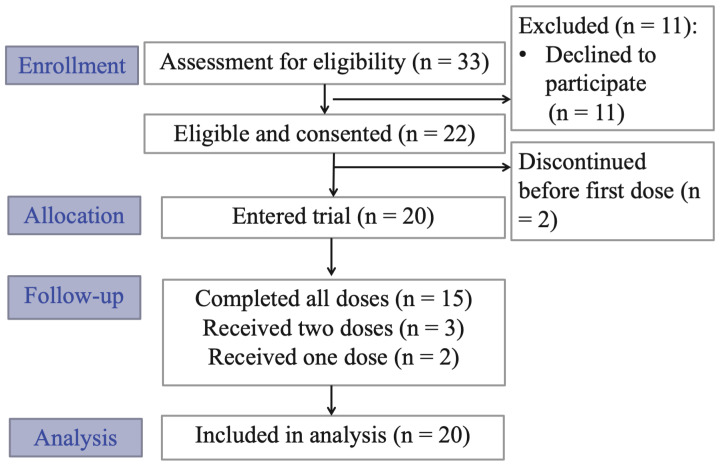
CONSORT clinical trial flow diagram.

**Figure 2 cancers-15-00400-f002:**
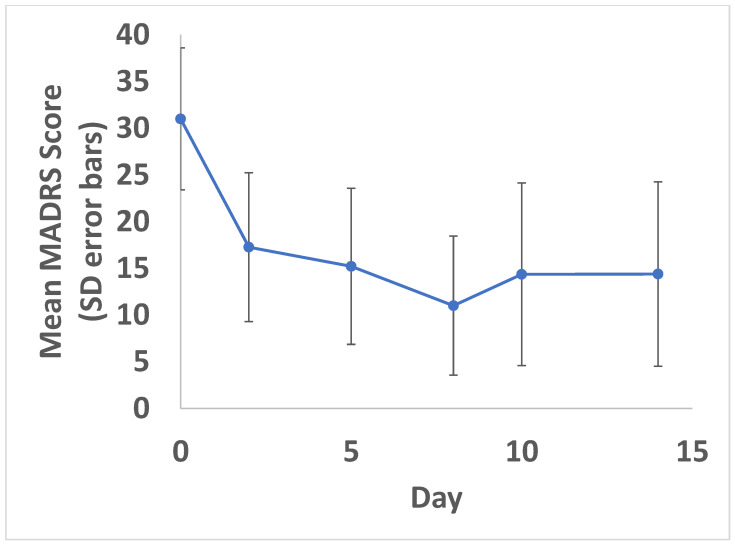
Change in MADRS over time. Mean MADRS scores with standard deviation (error bars) shown. A statistically significant reduction in symptoms was found at all time points compared to baseline (*p* < 0.05). Ketamine doses were administered on Days 1, 4 and 7.

**Figure 3 cancers-15-00400-f003:**
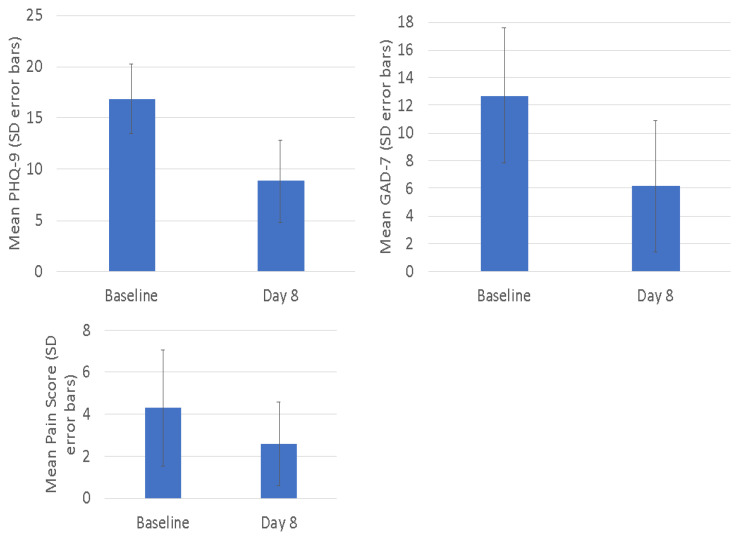
Secondary outcomes. Mean scores with standard deviation (error bars) shown.

**Table 1 cancers-15-00400-t001:** Baseline demographic information.

	Included Sample (n = 20)
**Age**	
**Mean (SD)**	58.4 (17.2)
**Range**	28–80
**Sex n (%)**	
**Female**	13 (65%)
**Male**	7 (35%)
**Marital status n (%)**	
**Married**	14 (70%)
**Single**	6 (30%)
**Baseline MADRS**	
**Mean (SD)**	31 (7.6)
**Range**	20–47
**Baseline PHQ-9 (SD)**	16.9 (3.3)
**Baseline GAD-7 (SD)**	12.7 (6.2)
**Life expectancy**	Range: 1–12 months
	Mean: 4 months
**Baseline psychiatric medications n (%)**	
**One antidepressant**	12 (60%)
**Two antidepressants**	3 (15%)
**Antipsychotics**	5 (25%)
**Psychostimulants**	4 (20%)
**Sleep aids**	3 (15%)
**Benzodiazepines**	9 (45%)
**No psychotropics**	2 (10%)
**Cancer type n (%)**	3 (15%)
**Pancreatic**	2 (10%)
**Ovarian**	1 (5%)
**Adrenal**	1 (5%)
**Rhabdomyosarcoma**	2 (10%)
**Prostate**	2 (10%)
**Breast**	1 (5%)
**Lymphoma**	3 (15%)
**Lung**	2 (10%)
**Myeloma**	1 (5%)
**CNS**	1 (5%)
**Colorectal**	1 (5%)
**Unknown primary**	

**Table 2 cancers-15-00400-t002:** Adverse events reported with systematic assessment using the ctcae v4.0.

Adverse Event	Frequency—n (%)
Dry mouth	2 (10%)
Dizziness	6 (30%)
Mild hypertension	3 (15%)
Somnolence/Fatigue/Drowsiness	5 (25%)
Vivid dreams	2 (10%)
Nausea	4 (20%)
Dysgeusia	10 (50%)
Panic attack	1 (5%)
Confusion	1 (5%)
Dissociation	6 (30%)
Headache	4 (20%)

## Data Availability

Data will not be made publicly available given that this is a small pilot study that did not obtain consent for making data publicly available.

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
