# Peer review of "A Phase II, Open-Label Clinical Trial of Intranasal Ketamine for Depression in Patients with Cancer Receiving Palliative Care (INKeD-PC Study)"

_cancers, 2023, doi:10.3390/cancers15020400_

Round 1

Reviewer 1 Report

Overall the study is great and has much merit to the field. 

For methods:

1) please provide more detail as to how tolerability of ketamine was used to determine dosage of 2nd and 3rd dose. Providing an example would be useful. For example: patients describing minimal side effects after dose 1 were given 100mg at dose 2, those with moderate side effects were given 75mg. 

2) Please use methods or results to describe the patients more. What were the types of cancers. How many were on other psychotropics? How did comorbid pain, insomnia impact the results. 

Results:

Under "safety and adverse events," please describe why 3 did not get 3 doses and what happened to the second participant who received only 1. The paragraph states 1 dropped after dose 1 due to side effects, what about the second? 

Discussion/conclusion: 

In the limitations, it is VITAL to describe that despite the promising results, ketamine should not be considered a replacement for mental health support at the EOL. A very brief reminder about the efficacy of psychological accompaniment at the end of life such as legacy building. 

How does ketamine compare to psilocybin for this? Would be good to review this in literature.

Author Response

Thank you for your review of our paper. We have made the suggested changes and believe it is a stronger paper now. Please find our itemized response herein. 

Reviewer 1:

Overall the study is great and has much merit to the field.

For methods:

1) please provide more detail as to how tolerability of ketamine was used to determine dosage of 2nd and 3rd dose. Providing an example would be useful. For example: patients describing minimal side effects after dose 1 were given 100mg at dose 2, those with moderate side effects were given 75mg.

Author’s Reply: We have added additional details as suggested on page 3 of the manuscript.

2) Please use methods or results to describe the patients more. What were the types of cancers. How many were on other psychotropics? How did comorbid pain, insomnia impact the results.

Author’s Reply: We have added information regarding cancer diagnosis and baseline psychiatric medications to Table 1. We agree these are important additions to better understand our sample. As shown, the sample had diverse cancer diagnoses and in general were prescribed antidepressants and other psychiatric medications at baseline.

We have not conducted the additional analysis of insomnia and pain as mediating/moderating factors. While this is an interesting research questions, we are unfortunately underpowered given the small sample size and believe that any results would be invalid and could be misleading (regardless of if there was a positive, negative or neutral impact on antidepressant effects). We do show in Fig. 3 that baseline pain scores were relatively low in this population and not significantly changed by ketamine.

Results:

Under "safety and adverse events," please describe why 3 did not get 3 doses and what happened to the second participant who received only 1. The paragraph states 1 dropped after dose 1 due to side effects, what about the second?

Author’s Reply: We have clarified “One dropout was related to adverse effects of ketamine. The other four dropouts were unrelated to study participation, but were related to changes in medical status (i.e., discharge from hospital or progression of cancer complications preventing participation).”

Discussion/conclusion:

In the limitations, it is VITAL to describe that despite the promising results, ketamine should not be considered a replacement for mental health support at the EOL. A very brief reminder about the efficacy of psychological accompaniment at the end of life such as legacy building.

Author’s Reply: We strongly agree with the reviewers comment regarding the importance of psychological support and agree that ketamine is no replacement for this. We have added “Given these important limitations, despite our promising results, we believe that ketamine should not be considered a replacement for conventional treatment and psychological support. Indeed, the importance of psychological interventions in advanced cancer, palliative care and near the end of life is strongly emphasized as a vital part of cancer care [4].”

How does ketamine compare to psilocybin for this? Would be good to review this in literature.

Author’s Reply: We have added this reference and sentence “Of note, our results were comparable to the very large effect sizes observed with psilocybin-assisted therapy for depression and anxiety in cancer, palliative and end of life populations [42].”

Reviewer 2 Report

I went through the manuscript two times and I did not find any crucial errors, minor editorial ones:

1. Please edit your affiliations, if you come from the same unit then you don't have to use a separate affiliations for each author.

2. I am not sure if MDPI allows "simple summaries" in their journals, therefore, you might want to just erase it, however it is entirely up to the editors.

3. Lines 74-75 - ketamine is not well estabilished to use in depression, the correction here is a must. What your probably meant was esketamine (Spravato by Janssen), which is registered to use in drug resistant MDD treatment. Racemic ketamine is used, but it is used off label and as far as I know it isn't registered for depression treatment anywhere in the world. This actually is beneficial for you, makes your CT way more important.

I don't have any more comments. The methodology applied is sufficient, the discussion is well prepared and in general, this is really good manuscript that deserves quick publication.

PS. As a psychiatrist myself, I like your study very much. Congratulations:)

Author Response

Thank you for your review of our paper. We have made the suggested changes and believe it is a stronger paper now. Please find our itemized response herein. 

Reviewer 2:

I went through the manuscript two times and I did not find any crucial errors, minor editorial ones:

  1. Please edit your affiliations, if you come from the same unit then you don't have to use a separate affiliations for each author.
  2. I am not sure if MDPI allows "simple summaries" in their journals, therefore, you might want to just erase it, however it is entirely up to the editors.

Author’s Reply: We believe this is the MDPI formatting requirements but are happy to change if needed based on editorial direction.

  1. Lines 74-75 - ketamine is not well established to use in depression, the correction here is a must. What your probably meant was esketamine (Spravato by Janssen), which is registered to use in drug resistant MDD treatment. Racemic ketamine is used, but it is used off label and as far as I know it isn't registered for depression treatment anywhere in the world. This actually is beneficial for you, makes your CT way more important.

Author’s Reply: Thank you for this important point. We have updated accordingly. We have reworded to say “extensively studied” rather than “well established.” We have also added “Of note, racemic ketamine is an off-label treatment for MDD, while esketamine nasal spray is FDA-approved as an add-on treatment for treatment resistant MDD.” Importantly, while racemic ketamine has not been approved for depression, it has still been extensively studied with >20 positive RCTs to date (see ref 14 and 27). As such, we believe that saying ‘extensively studied’ is reasonable for racemic ketamine, while acknowledging it is still off label.

I don't have any more comments. The methodology applied is sufficient, the discussion is well prepared and in general, this is really good manuscript that deserves quick publication.

  1. As a psychiatrist myself, I like your study very much. Congratulations:)

Author’s Reply: Many thanks for your review and kind words.